# Coronavirus M Protein Trafficking in Epithelial Cells Utilizes a Myosin Vb Splice Variant and Rab10

**DOI:** 10.3390/cells13020126

**Published:** 2024-01-10

**Authors:** Lynne A. Lapierre, Joseph T. Roland, Elizabeth H. Manning, Catherine Caldwell, Honor L. Glenn, Pierre-Olivier Vidalain, Frederic Tangy, Brenda G. Hogue, C. A. M. de Haan, James R. Goldenring

**Affiliations:** 1Department of Surgery, Vanderbilt University School of Medicine, Nashville, TN 37232, USA; lynne.lapierre@vumc.org (L.A.L.); joseph.t.roland@vumc.org (J.T.R.); elizabeth.h.manning@vumc.org (E.H.M.); catherine.caldwell@vumc.org (C.C.); 2Epithelial Biology Center, Vanderbilt University School of Medicine, Nashville, TN 37232, USA; 3Nashville VA Medical Center, Nashville, TN 37212, USA; 4Biodesign Institute Center for Immunotherapy, Vaccines & Virotherapy, Tempe, AZ 85287, USA; honor.glenn@asu.edu (H.L.G.); brenda.hogue@asu.edu (B.G.H.); 5Equipe Infections Virales, Métabolisme et Immunité, Centre International de Recherche en Infectiologie (CIRI), Univ. Lyon, INSERM U1111, CNRS UMR5308, Ecole Normale Supérieure de Lyon, Université Claude Bernard Lyon 1, 69008 Lyon, France; pierre-olivier.vidalain@pasteur.fr; 6Unité Génomique Virale et Vaccination, Institut Pasteur, CNRS UMR3569, 75015 Paris, France; 7Viral Genomics and Vaccination Unit, Department of Virology, Institut Pasteur, CNRS UMR3569, 75015 Paris, France; frederic.tangy@pasteur.fr; 8Center for Applied Structural Discovery, Biodesign Institute, Tempe, AZ 85287, USA; 9School of Life Sciences, Arizona State University, Phoenix, AZ 85004, USA; 10Faculty of Veterinary Medicine, Department of Biomolecular Health Sciences, Division of Infectious Diseases and Immunology, Section Virology, University of Utrecht, 3584 CS Utrecht, The Netherlands; c.a.m.dehaan@uu.nl; 11Cell and Developmental Biology, Vanderbilt University School of Medicine, Nashville, TN 37232, USA

**Keywords:** coronavirus, M protein, Rab10, Myosin Vb, MYO5B, Rab11a, membrane recycling, MHV

## Abstract

The membrane (M) glycoprotein of coronaviruses (CoVs) serves as the nidus for virion assembly. Using a yeast two-hybrid screen, we identified the interaction of the cytosolic tail of Murine Hepatitis Virus (MHV-CoV) M protein with Myosin Vb (MYO5B), specifically with the alternative splice variant of cellular MYO5B including exon D (MYO5B+D), which mediates interaction with Rab10. When co-expressed in human lung epithelial A549 and canine kidney epithelial MDCK cells, MYO5B+D co-localized with the MHV-CoV M protein, as well as with the M proteins from Porcine Epidemic Diarrhea Virus (PEDV-CoV), Middle East Respiratory Syndrome (MERS-CoV) and Severe Acute Respiratory Syndrome 2 (SARS-CoV-2). Co-expressed M proteins and MYO5B+D co-localized with endogenous Rab10 and Rab11a. We identified point mutations in MHV-CoV M that blocked the interaction with MYO5B+D in yeast 2-hybrid assays. One of these point mutations (E121K) was previously shown to block MHV-CoV virion assembly and its interaction with MYO5B+D. The E to K mutation at homologous positions in PEDV-CoV, MERS-CoV and SARS-CoV-2 M proteins also blocked colocalization with MYO5B+D. The knockdown of Rab10 blocked the co-localization of M proteins with MYO5B+D and was rescued by re-expression of CFP-Rab10. Our results suggest that CoV M proteins traffic through Rab10-containing systems, in association with MYO5B+D.

## 1. Introduction

The *Coronaviridae* family includes multiple animal and human (hCoV) pathogenic viruses, including MHV-CoV, PEDV-CoV, MERS-CoV, SARS-CoV and the COVID-19 pandemic virus SARS-CoV-2, as well as endemic less pathogenic human respiratory viruses OC43-CoV, 229E-CoV, NL63-CoV and HKU1-CoV [1]. Coronaviruses are assembled from four primary structural proteins encoded in the viral RNA genome [1]. Among these proteins, the membrane (M) glycoprotein is the major component of the virus envelope that plays a major role in virus assembly through its interactions with the envelope (E), nucleocapsid (N) and spike (S) viral proteins [2,3,4]. CoV M proteins are glycosylated at the amino-terminus through either N- or O-linkage. Members of the alpha genus are glycosylated by N-linkage at asparagine residues, whereas the beta-coronavirus M protein exhibits O-linked glycosylation at serine and threonine residues [5].

M proteins traffic to the Golgi apparatus, and previous studies have indicated that virion particles are assembled in the endoplasmic reticulum–Golgi intermediate compartment (ERGIC) and egress through the cellular exocytic pathway [6,7]. Recent investigations have suggested that coronavirus particles may also be released through exocytotic lysosomes [8]. Although earlier studies presented an order of virion assembly, few investigations have addressed interactions of virion proteins with components of the vesicle trafficking systems in epithelial host cells.

Polarized epithelial cells maintain distinct apical and basolateral domains by segregating vesicle trafficking pathways to deliver cargoes to specific cell surfaces [9,10,11]. In particular, the apical recycling system is responsible for trafficking proteins internalized from the apical membrane back to the apical membrane (apical recycling). The apical recycling system also mediates basolateral to apical transcytosis. The apical recycling system therefore acts as a critical processing center in polarized cells to coordinate trafficking towards the apical membrane. While Rab11a is most associated with the apical recycling system, Rab25, Rab8a and Rab10 have all been associated with aspects of membrane recycling [11,12]. All of these small GTPases share the ability to interact with class V myosin motors, in particular Myosin Vb (MYO5B) [13,14,15]. Rab11a, Rab11b, Rab25 and Rab8a appear to interact with all of the splice variants of MYO5B, but Rab10 only interacts with the relatively rare MYO5B splice variant that contains the alternatively spliced 26 amino acid exon D sequence [14]. The MYO5B splice variant without exon D (MYO5BΔD) is ubiquitous, but the MYO5B variant with exon D (MYO5B+D) is enriched in the brain, lung, heart and intestine [14].

We have investigated the association of M protein with host regulators of membrane trafficking. We identified an interaction between coronavirus M proteins with the specific splice variant of host myosin Vb (MYO5B) that harbors the 26-amino acid alternatively spliced exon D (MYO5B+D) sequence. The carboxyl-tail of MHV M protein, located in the cytoplasm during assembly, interacted only with MYO5B constructs containing exon D, which is also responsible for interactions of the motor with the small GTPase, Rab10. Specific mutations in the MHV-CoV M tail blocked interactions with MYO5B+D. One of the mutants, E121K, has been previously shown to block the assembly of MHV-CoV virions [16]. Loss of Rab10 also abrogated the interaction between M proteins and MYO5B+D, and re-expression of Rab10 re-established the co-localization of M proteins with MYO5B+D. This newly identified interaction of coronavirus M glycoproteins with MYO5B+D suggests that M proteins traffic through a Rab10-dependent vesicle trafficking pathway.

## 2. Results

A yeast 2-hybrid screen was performed with the cytoplasmic tail of MHV M protein as a bait to identify interacting proteins in a mouse brain prey library. We identified an interaction between the MHV M protein tail and Myosin Vb (MYO5B) (Figure 1A). Sixty clones of Myo5b, beginning with amino acids 997 to 1249 and extending to the carboxyl terminal end of the protein, were identified. Using a series of 2-hybrid assays, we subsequently confirmed the association and demonstrated that the M cytoplasmic tail interacted with one specific alternatively spliced exon of Myo5b: exon D (Figure 1A). Interestingly, the MHV M tail did not interact with MYO5B sequences lacking exon D (Figure 1A). A construct containing just the ABCDE exons interacted with MHV M, but the ABCE exon sequence did not (Figure 1A). In two-hybrid assays, MHV M tail protein did not interact with either Myosin Va or Myosin Vc sequences, even though they did contain appropriate D exon sequences (Figure 1A). We previously reported that Rab10 interacts with the alternatively spliced exon D in both MYO5A and MYO5B and in a homologous unspliced region in MYO5C [14]. We determined that, while the splice variant of MYO5B without exon D is ubiquitous, the splice variant with exon D is most highly expressed in the lung, brain, heart and intestines, prominent tissue targets of coronavirus infection [14]. All of these findings led to the conclusion that MHV M protein forms a specific association with MYO5B containing Exon D.

We next utilized the yeast 2-hybrid assay between the MHV M tail and MYO5B (ABCDE) exons to evaluate whether truncations of the M carboxyl terminus would alter the interaction (Figure 1B). Previous studies had shown that deletion or substitutions of the terminal 2 amino acids of the M tail would impair MHV assembly [17,18,19]. Interestingly, truncation of the last two amino acids, as well as the last twenty amino acids, had no effect on the interaction between MYO5B(ABCDE) and the MHV M tail (Figure 1B). The latter truncation resulted in the elimination of the conserved region, which has been reported as important for Golgi trafficking of the MERS M protein [20]. The insertion of triple alanine substitutions (ANAA) in the domain did not affect the MHV M tail interaction with MYO5B(ABCDE). However, the extension of the truncation to 40 carboxyl-terminal amino acids abrogated the interaction (Figure 1B). To provide greater insights into the requirements for the MYO5B-M tail interaction, we performed a random mutagenesis of the MHV M cytosolic tail. The mutated sequences were ligated back into the pBD-Gal-Cam bait vector and screened in a yeast two-hybrid assay against yeast with pAD-MYO5B (ABCDE) (Figure 1B). We identified seven point-mutations in the MHV M tail which disrupted the interaction with MYO5B(ABCDE). These point mutations spanned a 50-amino acid length within the M tail sequence. Interestingly, we previously showed that the E121K mutation in the MHV M protein blocked the assembly of both virus-like particles (VLPs) and virions [16]. The residue is located in the conserved domain following the third transmembrane in all coronavirus M proteins [16]. Altogether, the results of these yeast two-hybrid assays indicate that the M protein carboxyl tail interacts with the MYO5B exon D splice variant and that the integrity of the tail is important for this interaction.

To understand further the interaction of MYO5B+D with the MHV M tail, we extended the study to examine the interaction in cultured cells. The full-length M protein was tagged with EGFP at the carboxyl end of the tail and transiently expressed in MDCK cells. When MHV M-GFP was expressed alone in MDCK cells, it localized predominantly to lateral and junctional membranes, with some intracellular perinuclear staining (Figure 1C). When MHV M-GFP and Cherry- MYO5B+D full length proteins were co-expressed in MDCK cells, they colocalized in membrane puncta that also contained both Rab11a and Rab10 (Figure 1C,D). Cherry- MYO5B+D and MHV M-GFP did not colocalize with Rab8a (Appendix A). The membranes harboring MHV M-GFP and Cherry- MYO5B+D were distinct from the Golgi membranes, labelled by GM130 immunostaining (Appendix A), and were consistently associated with an accumulation of F-actin (Figure 1C). Importantly, when Cherry-MYO5B without the D exon (MYO5BΔD) was co-expressed with MHV M-GFP, little co-localization was observed, and more of the M protein was seen along the lateral and junctional membranes (Figure 1C). The analysis of Manders’ coefficients demonstrated a significant loss in co-localization (Figure 2A).

Given that the MHV M(E121K) mutant blocked the interaction with MYO5B+D in yeast two-hybrid assays, we further evaluated the effects of the mutation in MHV M in MDCK cells. Figure 3 demonstrates that the E121K point mutation caused a loss of co-localization with MYO5B+D (Figure 2A) and Rab10. These findings suggest that a point mutation that alters MHV M interactions with MYO5B can affect trafficking of the viral M glycoproteins.

We chose to utilize MDCK cells because coronaviruses normally infect polarized epithelial cells at points of primary entry in multiple organ systems, especially the gut and lungs. Additionally, MDCK cells model the specialization of the apical recycling endosome system, which is enriched for Rab11a and MYO5B [11,13]. To determine if the localization findings were particular to MDCK cells, we also evaluated colocalization in A549 lung epithelial cells. Appendix A demonstrates that, similar to the MDCK cells, MHV M-GFP colocalized with Cherry- MYO5B+D. The M protein co-localized with Rab10, and little localization in the Golgi apparatus was observed. Also, as in the MDCK cells, the E121K mutation in MHV M-GFP blocked co-localization with Cherry-M5B + D when co-expressed in A549 cells (Appendix A). Previous investigations have also suggested that coronavirus particles were assembled in the ERGIC [21]. While no antibodies are available to establish the distribution of ERGIC in MDCK cells, we were able to assess the localization of MHV M-GFP in the A549 cells. Little overlap with ERGIC-53 staining was observed (Appendix A).

In classical yeast 2-hybrid screens, we were not able to definitively identify interactions of coronavirus M tails from PEDV, MERS or SARS-CoV-2 with MYO5B(ABCDE). Nevertheless, previous studies have demonstrated that the cytosolic tails of M proteins have close interactions with the cytosolic leaflet of membranes [17]. We therefore examined the interactions of coronavirus M proteins with MYO5B exons ABCDE using a split-ubiquitin yeast 2-hybrid assay that assesses interactions at membrane surfaces. M proteins from MHV, PEDV, MERS and SARS-CoV-2 all interacted with MYO5B exons ABCDE in the split-ubiquitin assay (Table 1). These findings are compatible with the concept that the interaction of the M protein tails with the membrane inner leaflet may promote their presentation, facilitating interaction with MYO5B+D.

Based on the findings in the split-ubiquitin yeast 2-hybrid studies, we next assessed the localization of M-GFP chimeric proteins for PEDV, MERS and SARS-CoV-2 co-expressed with Cherry- MYO5B+D in MDCK cells (Figure 4). In all three cases, we observed colocalization between the M-GFP chimeric proteins and co-expressed Cherry- MYO5B+D in an internal vesicular compartment. In addition, Manders’ coefficients demonstrated that co-localization was abrogated when M proteins were expressed with Cherry-MYO5BΔD (Figure 2B, Figure 4). We observed similar results for co-expression of coronavirus M-GFP protein chimeras with MYO5B+D in A549 lung cells (Appendix A). All coronavirus M proteins contain a glutamate residue in a homologous position to the E121 residue in MHV M [16]. We observed that the mutation of this glutamate residue to lysine in all of the coronavirus M proteins abolished co-localization with MYO5B+D (Figure 2C and Figure 5) in MDCK cells. Similar results were also found with the coronavirus M protein E to K mutants expressed in A549 cells (Appendix A).

Our previous studies have shown that the D exon of MYO5B is responsible for interactions with Rab10 [14]. A motorless tail construct of MYO5B can cause concentration of MYO5B with binding Rab proteins in collapsed membranous cisternae [13]. We therefore examined whether the knockdown of Rab10 could alter the trafficking of coronavirus M proteins. We made an inducible Rab10 knockdown in A549 cells. The mean intensity of the Rab10 band in the Rab10 shRNA cells was 0.265 ± 0.0444 compared to a mean intensity of 1 for cells expressing the non-targeting (NT) shRNA (Figure 6A). Figure 6B demonstrates that the expression of an mCherry- MYO5B+D tail construct (902T) in the Rab10KD A549 cells not treated with doxycycline caused the concentration of MHV M-GFP protein with endogenous Rab10. Nevertheless, when the Rab10KD A549 cells were treated with doxycycline, leading to the knockdown of Rab10, we observed a loss of colocalization of MHV M-GFP with MYO5B+D tail (Figure 6B and Figure 7). Reintroduction of CFP-Rab10 resistant to shRNA, re-established the colocalization of MYO5B+D tail with MHV-M in the doxycycline treated Rab10 KD cells. Appendix A demonstrates that a similar sequestration of M proteins was observed for M proteins from MERS, PEDV and SARS-CoV2 with expression of a mCherry- MYO5B+D tail. As observed with MHV M, induced knockdown of Rab10 expression led to a loss of colocalization of MYO5B+D with M proteins (Figure 7 and Appendix A). Re-expression of CFP-Rab10 re-established colocalization of M proteins with MYO5B+D tail (Figure 7).

## 3. Discussion

Taken together, our results indicate that the relatively rare MYO5B+D splice variant associates with the cytoplasmic tail of coronavirus M proteins when the proteins are co-expressed in polarized cells. Previous investigations have indicated that MYO5B is a multifunctional regulator of vesicle trafficking through its interactions with Rab11a, Rab11b, Rab25, Rab8a and Rab6a [15,22,23], but only the alternatively spliced exon D, with an additional 26 amino acids, can interact with Rab10 [14]. While the splice variant without exon D is ubiquitous, the expression of MYO5B+D is enriched in the lung, brain, heart and intestine, all target tissues for coronavirus infection [14]. We and others have noted previously that other viruses, including respiratory syncytial virus [24,25,26], influenza [27], Epstein Barr Virus [28], Ebola [29], and HIV [30,31] utilize component regulators of the plasma membrane/apical membrane recycling systems to accomplish viral protein trafficking in host cells. A recent bioinformatics investigation predicted that Myosin V motors might be involved in coronavirus trafficking through interactions with non-structural coronavirus proteins [32]. Our data suggest that, in epithelial cells, a post-Golgi, Rab10-containing membrane trafficking compartment may be critical for M protein trafficking. M proteins accumulated with MYO5B+D in vesicles containing Rab11a and Rab10. Rab11a is strongly associated with the apical recycling system in polarized cells [11,12], and Rab10 has been associated with both endosomal and exocytotic pathways in different cell systems [33,34]. Still, Rab10-specific cargoes and vesicle trafficking pathways have not been identified. Much progress has been made in examining vesicle trafficking systems through the study of viral protein trafficking in the endoplasmic reticulum and Golgi apparatus [35,36]. Similarly, coronavirus M-GFP chimeric proteins represent novel cargoes for the examination of Rab10-dependent trafficking systems.

While previous studies had suggested that coronavirus virions are assembled in the ERGIC and egress through the exocytic pathway [7], a recent report proposed that MHV virions can also be released from secretory lysosomes in HeLa cells [8]. Our present findings suggest that, in epithelial cells, elements of the apical recycling system may play a role in M protein trafficking. Our previous investigations have demonstrated that MYO5B can act as a processive anchor, retarding the movement of Rab11a-containing vesicles along microtubule tracks [37]. Similarly, MYO5B+D may serve as a break on the normal forward membrane protein trafficking through the recycling system. Whether final egress can be accomplished through recycling endosomes, Rab10-positive post-Golgi vesicles acting alone or through fusion with secretory lysosomes remains to be fully understood.

## 4. Conclusions

In summary, our studies have identified MYO5B+D as a host protein that interacts with coronavirus M proteins. We have identified the ability of an E121K mutation in MHV M to disrupt binding to MYO5B+D. Since our previous studies showed this same mutation blocks the assembly of both virus-like particles and virion assembly [16], it seems reasonable to suggest that MYO5B+D may play a role in M protein trafficking in host cells. Since the association between M proteins and MYO5B+D appears to be conserved among coronavirus M proteins, these findings suggest that coronavirus M protein trafficking may be an important assay for the elucidation of Rab10-mediated membrane dynamics.

## 5. Materials and Methods

### 5.1. DNA Sequence Construction

DNA sequences for full length and cytoplasmic tails of M proteins PEDV, MERS and SARS-CoV-2 were synthesized with appropriate flanking restriction sites by GeneArt (Thermo Fisher, Waltham, MA, USA). M protein cytoplasmic tails were ligated into pBD-Gal(Cam) (Stratagene, La Jolla, CA, USA) following restriction digest with EcoRI and SalI and gel isolation. Full-length M protein sequences were ligated in pEGFP-N1 vector (Clontech, San Francisco, CA, USA) following restriction digest with EcoRI and SalI and gel isolation. Site-directed mutagenesis of M protein sequences was performed in all cases using a single nucleotide mutagenesis protocol (see Appendix A for sequences) utilizing the Quickchange Lightening Multi Site-directed Mutagenesis Kit (Agilent, Santa Clara, CA, USA). All mutagenized sequences were confirmed using Sanger sequencing (GenHunter, Nashville, TN, USA). The sequences of human MYO5B+D 902 tail and MYO5BΔD 902 tail were inserted into pmCherry-C1 vector (Clontech) using flanking BamHI and SalI restriction sites.

### 5.2. Yeast Two-Hybrid (Y2H) Screen

The Y2H screen was performed following the protocol described in Vidalain et al. [38]. The DNA sequence encoding the C-terminal cytoplasmic tail of membrane protein (Mcyto) of mouse hepatitis coronavirus strain A59 (starting with S105) was cloned via in vitro recombination (Gateway technology; Invitrogen, Carlsbad, CA, USA) from pDONR207 into the Y2H vector pPC97-GW (kindly provided by Dr. Marc Vidal) for expression in fusion downstream of the GAL4 DNA-binding domain (GAL4-BD). AH109 yeast cells (Clontech; Takara, Mountain View, CA, USA) were transformed with the GAL4-BD-Mcyto plasmid using a standard lithium acetate protocol. Transactivation of the HIS3 reporter gene by the GAL4-BD-Mcyto protein alone was tested on culture medium lacking histidine, and the concentration of 3-amino-1, 2, 4-triazole (3-AT) was adjusted to prevent spontaneous yeast growth in the absence of interacting prey protein during the screen. In parallel, a mouse brain cDNA library cloned into yeast two-hybrid vector pPC86 (Life Technologies, Carlsbad, CA, USA) to express prey proteins in fusion downstream of the GAL4 transactivation domain (GAL4-AD) was transformed into Y187 yeast cells (Clontech; Takara, Mountain View, CA, USA). Finally, AH109 cells expressing GAL4-BD-Mcyto were mated with Y187 cells expressing the prey cDNA library and were plated on -L-T-H+3AT selective medium. After six days of culture, colonies were picked, and replica-plated over three weeks on selective medium to eliminate the potential for contamination with false positives. Prey proteins from selected yeast colonies were identified via PCR amplification using primers that hybridized within the pPC86 regions flanking the cDNA inserts. PCR products were sequenced, and cellular interactors were identified through BLAST analyses.

### 5.3. Binary Yeast Two-Hybrid Assays

The DNA sequences to be screened were cloned into either the pAD activation vector or the pBD bait vector (Clontech). For yeast 2-hybrid assays, a culture of Y190 yeast was grown overnight and diluted to OD_600_ = 0.2 in 2 mL YPDA media per reaction, then allowed to grow to log phase. Yeast cells were pelleted at 3000× *g* for 3 min. The cells were resuspended in 20 μL per sample of 0.1 M lithium acetate in TE buffer (10 mM Tris-HCl, 1 mM EDTA, pH 7.5). This solution was added to 120 μL per sample of 0.1 M lithium acetate/40% PEG 3350 in TE buffer and mixed by inversion. Each pAD and pBD vector was aliquoted to 1 μg, then 135 μL of transformation solution was added to the vectors for each reaction and incubated for 30 min at 30 °C with shaking at 75 rpm. DMSO was added to each reaction to 10% and reactions were incubated for 15 min at 42 °C. Yeast cells were pelleted at 2000× *g* for 3 min; the supernatant was removed, and the yeast cells were resuspended in 30–50 μL H_2_O and plated on Trp/Leu drop-out plates. After 3 days of growing at 30 °C, a colony lift assay was performed. Filter paper was pressed onto colonies to lift them from the plate, then the filter paper was frozen for 20 s in liquid nitrogen. The filter papers were thawed for several minutes, then frozen in liquid nitrogen again for 20 s. After several minutes of thawing, 2 mL 0.33 mg/mL X-gal (from 20 mg/mL solution in DMF) in Z-buffer (60 mM Na_2_HPO_4_-7H_2_O, 40 mM NaH_2_PO_4_-H_2_O, 10 mM KCl, 1 mM MgSO_4_-7H_2_O, pH 7) was dispensed onto a clean filter paper, then each filter paper with frozen colonies was placed onto a wet filter paper to absorb the X-gal solution. Colony color change was observed every 30 min for approximately 3 h. All binary yeast 2-hybrid assays were repeated at least 3 times to identify consistent results.

### 5.4. Random Mutagenesis

For the random mutagenesis screen, a PCR reaction was performed on selected DNA inserts in pBD vectors using Qiagen Taq polymerase (201203). The reaction was composed of 0.2 μg pBD vector with MHV M cytoplasmic tail insert, 1× Qiagen PCR buffer (201203), 0.2 mM dNTPs, 1 μM each sense and anti-sense primer, and 1% Taq polymerase with a total volume of 100 μL. The reaction was heated for 5 min at 94 °C, then run for 10 cycles of 30 s at 94 °C, 30 s at 58 °C and 1 min per 1000 base pairs at 72 °C, then heated for 5 min at 72 °C. MnCl_2_ was added to the reaction to 5 μM, then the reaction was run for 30 cycles of 30 s at 94 °C, 30 s at 58 °C and 2 min at 72 °C, then heated for 5 min at 72 °C. The PCR product was purified using the QiaQuick PCR Purification Kit (28104), the mutated insert was cut out of the pBD vector, gel-isolated, extracted using the QiaQuick Gel Extraction Kit (28704) and ligated into an empty pBD vector.

The pBD vector library with a mutagenized MHV M cytoplasmic tail insert was screened against the target pAD-MYO5B(ABCDE) vector by transforming the vectors into yeast, as described above, and performing a blue-white β-galactosidase assay. Colonies that stayed white in the blue-white assay were grown up in YPDA media for 3 days, then vectors were harvested using the QiaPrep Spin Miniprep Kit (27104) with the addition of 5 min of vortexing yeast with ~100 uL of glass beads after adding resuspension buffer. The resulting vectors were transformed into DH5α cells and grown on chloramphenicol plates to select for the pBD vectors with mutagenized inserts. Vectors were harvested from the bacterial colonies and sequenced using Sanger sequencing. All putative mutant sequences were confirmed with binary yeast 2-hybrid assays using rescued vector sequences.

### 5.5. Split-Ubiquitin Yeast 2-Hybrid Assays

DNA sequences for full length M proteins were cloned into the pCCW bait vector, while the MYO5B exons ABCDE sequences were cloned into the pDSL prey vector or the Dualsystems Biotech vector. For split-ubiquitin assays, a culture of Nym32 yeast was grown overnight in YPDA media and diluted to OD_600_ = 0.2 in 2 mL YPDA media per reaction, then allowed to grow to log phase. The yeast cells were pelleted at 3000× *g* for 3 min. The yeast cells were resuspended in 100 μL H_2_O per sample, then added to tubes containing 290 μL of transformation solution (5% salmon sperm DNA, previously heated to 100 °C/40% PEG 3350/0.12 M lithium acetate) and 1 μg of the appropriate pDSL and pCCW vectors for each transformation. The transformation reactions were mixed via vortexing for 20 s, then were incubated for 45 min in a 42 °C water bath. Yeast cells were pelleted at 2000× *g* for 3 min; the supernatant was removed, and the yeast cells were resuspended in 30–50 μL H_2_O and plated on -Trp/-Leu plates.

After 3 days of growing at 30 °C, a spot assay was performed using the transformed yeast. Yeast cells from each transformation were resuspended in 500 μL H_2_O and diluted to the same OD_600_ between 2 and 5, depending on each experiment. The yeast cells were then diluted 1:10, 1:50 and 1:100, and 5 μL of each dilution (including the original 1× dilution) were spotted onto -Trp/-Leu/-His and -Trp/-Leu/-His + 10 mM 3-amino-1,2,4-triazole (Sigma-Aldrich, St. Louis, MO, USA) plates, then air-dried. The yeast cells were grown for 3 days at 30 °C, then the plates were photographed.

Yeast growth was rated on a scale of 0–8 in which 0 was no growth, 1 was a few colonies in the 1× spot, 2 was a solid dot in the 1× spot, 3 was a solid dot in 1× and a few colonies in the 1:10 spot, 4 was solid dots on 1× and 1:10, and so forth for all 4 dilutions.

### 5.6. Cell Culture and Transfections

Madin-Darby Canine Kidney (MDCK) cells and the A549 human epithelial lung cell line [39] (ATCC) were plated onto coverslips or 6-well plates and transfected with the indicated vectors using PolyJet (SignaGen Labs, Frederick, MD, USA). For co-transfection in immunofluorescence, the ratio of vectors for mCherry N-terminally tagged MYO5Bs to the EGFP C-terminally tagged Coronavirus M proteins was 2:1.

### 5.7. Inducible Rab10 Knockdown Line

For the tetracycline-inducible Rab10 knockdown cell lines, the shRNA insert from a lentiviral pLKO.1 shRNA vector targeting human Rab10 (Sigma Aldrich TRCN0000382083) and the shRNA insert from a lentiviral pLKO.5 non-target shRNA control vector (Sigma-Aldrich SHC216-1EA) were cloned into tet-pLKO-puro (Addgene 21915) and used for lentiviral transduction of A549 cells (ATCC). HEK 293FT cells (ATCC) were plated on T75 flasks and grown to ~50% confluence. To transfect the HEK 293FT cells, 30 μL of Polyjet (SignaGen Labs) transfection reagent per flask was used with a mixture of DNA consisting of 4 μg of the Rab10 targeting or control vector, 4 μg of the packaging plasmid psPAX2 and 2 μg of the ENV plasmid pMD2.G. The HEK 293FT cells were refed after 24 h and incubated for an additional 48 h. The media from the cells were then filtered, incubated overnight with the addition of LentiX Concentrator (Takara, San Jose, CA, USA) 1:3, spun down for 45 min at 1500× *g*, resuspended in media and frozen until use. Concentrated virus was used with polybrene at 5 μg/mL to transduce plated A549 cells at 70% confluence. After 48 hrs, the A549 cells were selected in medium containing 4 μg/mL puromycin (Cellgro, Lincoln, NE, USA).

### 5.8. Western Blotting Analysis

Cells were grown on 12-well plates and treated daily for 72–96 h with doxycycline at 4 μg/mL, then washed in ice cold TBS, scraped, pelleted, resuspended in 40 μL of 65 °C 1% SDS/1 mM EDTA and sonicated twice for 15 s. After addition of Laemmli sample buffer, lysates were run on 4–12% polyacrylamide gels, then transferred onto Odyssey nitrocellulose membrane (LiCor, Lincoln, NE, USA) for 1 h at 75 V. Membranes were blocked with Odyssey TBS Intercept blocking buffer (LiCor), probed with rabbit anti-Rab10 (see Appendix A) and then probed with Odyssey anti-rabbit 800 nm secondary antibody (LiCor). Blots were imaged using a LiCor Odyssey Fc. Membranes were then stained with Ponceau S (Sigma, St. Louis, MO, USA) to visualize total protein, and colorimetric images were taken with an Amersham Imager 680. Densitometric analysis of Western blot bands for three experiments was performed using ImageJ (NIH). Protein levels were normalized to each lane’s corresponding Ponceau S signal. All data are presented as mean ± standard error of the mean (SEM), and statistically significant differences were computed in GraphPad Prism 10.1.1 (GraphPad Software, Boston, MA, USA) using Student’s *t*-test, the level being set at *p* < 0.05.

### 5.9. Immunofluorescence and Colocalization

Doxycycline was added to the inducible Rab10 knockdown line 2–4 h after transfection. The cells were grown with doxycycline (4 μg/mL) for 48–72 h, with the addition of fresh doxycycline every 24 h. Transfected cells were fixed in 4% paraformaldehyde for 20 min at room temperature (RT) then blocked and extracted in 10% normal donkey sera (Jackson ImmunoResearch, West Grove, PA, USA), 0.3% Triton X-100 PBS for 30 min at RT. For primary antibodies used, see Appendix A. All secondary antibodies used were from Jackson ImmunoResearch. The stained cells were mounted with ProLong Gold (Invitrogen, Carlsbad, CA, USA). The cells were imaged on a Zeiss LSM 710 or Zeiss LSM 980 confocal microscope using an 63x/1.40 Plan-Apochromat oil immersion lens and 2.5X zoom. Images were processed utilizing Zeiss ZEN blue software (version 3.8.99), and the figures were assembled in Adobe Photoshop. Co-localization analysis was performed on the images using Imaris software version 10.1 (Oxford Instruments, Santa Barbara, CA, USA). The raw.csz images were converted into the Imaris format with Imaris converter, then opened in the coloc module of Imaris. Thresholds for each image were set utilizing the 2D histogram for the GFP and Cherry channels. The ignore border bins function was activated to remove the 0 red and 0 green voxels from the image, removing a large number of non-fluorescent voxels. Thresholds were set to include only voxels along the diagonal of the 2D histogram, excluding the bright voxels directly along the *X* or *Y* axis. Once the thresholds were set, a 3D colocalization channel was created. The calculated data from this channel were exported into Excel, then imported into Prism to create Manders’ graphs. Differences in Manders’ coefficients were evaluated with a Student’s *t*-test. Differences in Pearson’s coefficients for Rab10 KD and rescue cells were evaluated using ANOVA with post hoc comparison of significant means with Tukey’s test. The Zeiss 710 and Zeiss 980 microscopes and Imaris software are maintained in the Vanderbilt Cell Imaging Shared Resource.

## Figures and Tables

**Figure 1 cells-13-00126-f001:**
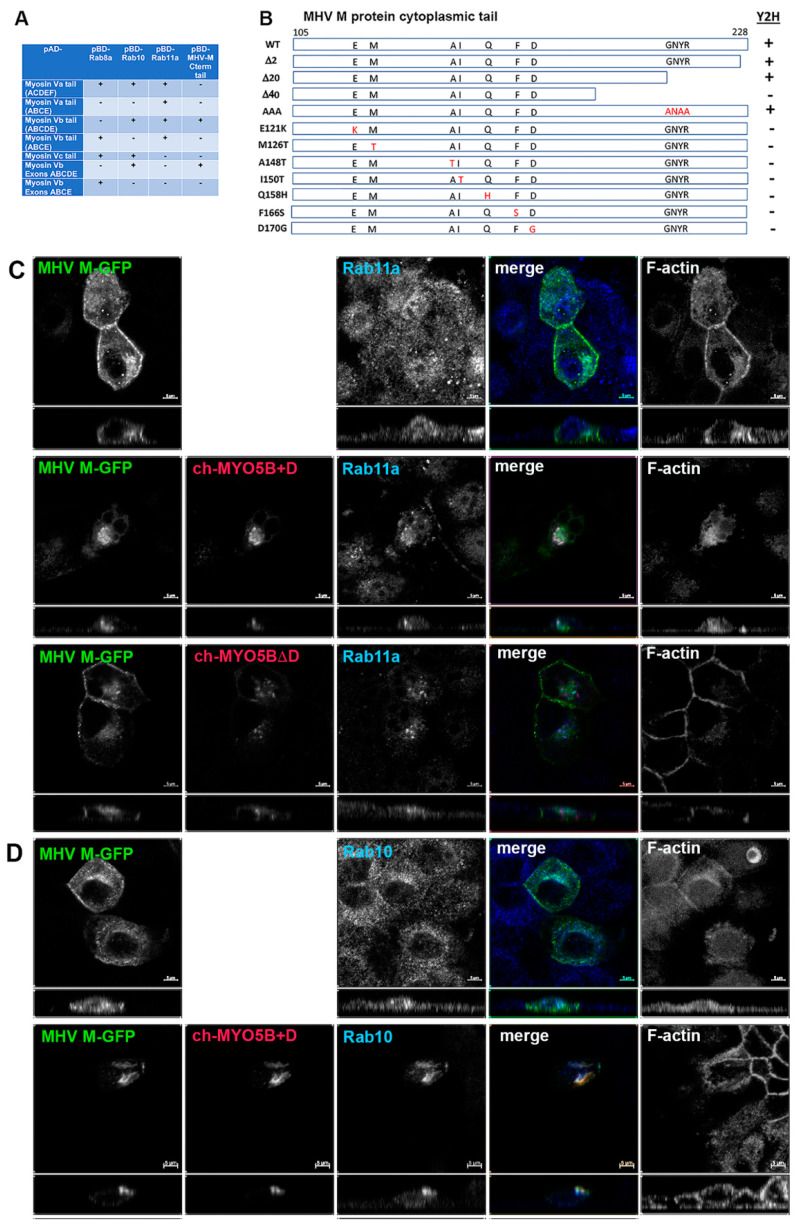
The MHV M cytoplasmic tail interacts with MYO5B+D. (**A**) Yeast 2-hybrid interactions between pAD-Myosin Vb targets and various interacting proteins, including Rab GTPases and Rab11-FIP2, which are known to interact with Myosin V proteins. Note that the MHV M cytoplasmic carboxy-terminal tail only interacts with Myosin Vb constructs that include the D exon. (**B**) Truncation and random mutagenesis reveal requisite binding regions in MHV M tail (amino acids 105–228). Truncations of MHV M cytoplasmic tail or random mutagenesis of pBD-MHV M tail were assayed in yeast 2-hybrid reactions with pAD-MYO5B(ABCDE). Yeast 2-hybrid results are noted on the right (Y2H). (**C**) Co-expression of MHV M-GFP with Cherry- MYO5B+D or Cherry MYO5BΔD and immunostaining of endogenous Rab11a and Phalloidin staining of F-actin in MDCK cells. Labels on the individual panels indicate the color used to produce the three-color merged image. Panels to the right of the merged panel were not used to produce the merged image. *Z* axis projections are shown below X-Y slice images, with the merged overlap at the right. (**D**) Co-expression of MHV M-GFP with Cherry MYO5B+D and endogenous Rab10 immunostaining (cyan) and Phalloidin staining of F-actin (blue) in MDCK cells. All results are representative of four separate experiments. Bar = 5 µm.

**Figure 2 cells-13-00126-f002:**
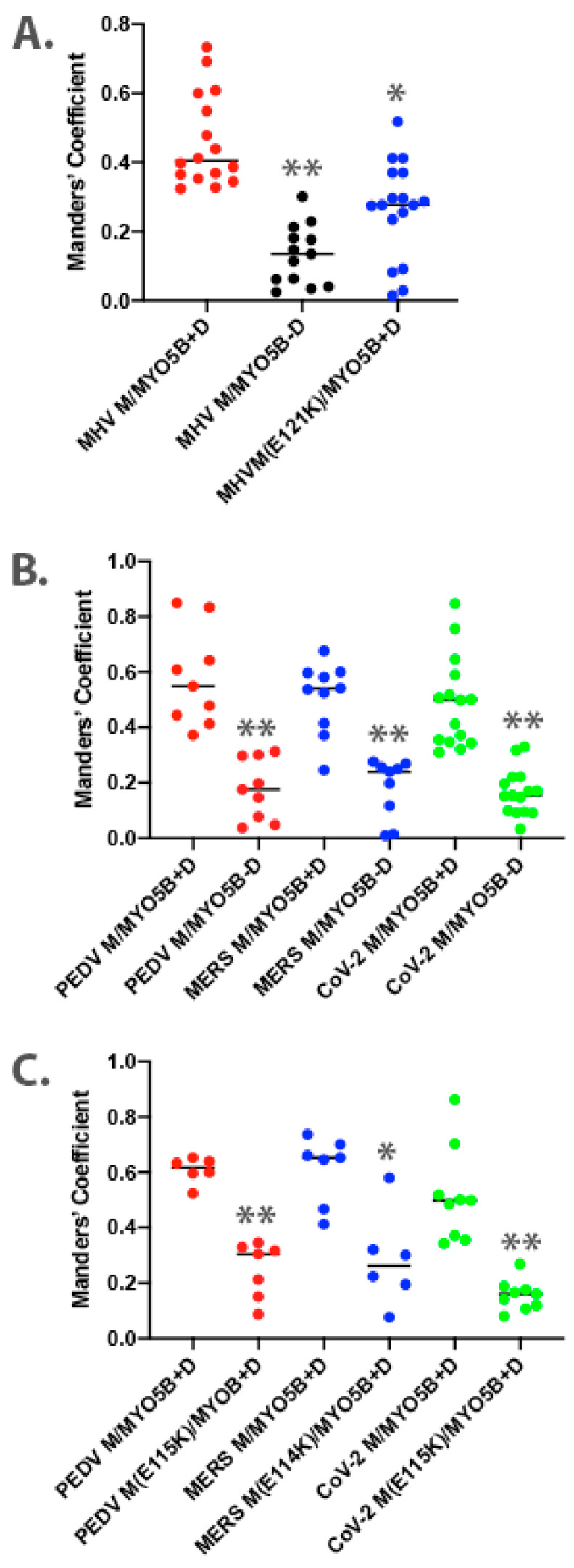
Colocalization analysis for dual expression of M proteins with MYO5B. Three-dimensional Manders’ coefficients were calculated for Z-stack images for dually transfected MDCK cells. Dual expression pairs are noted on the *X*-axis. (**A**) MHV M-GFP expression with MYO5B+D or MYO5BΔD (MYO5B-D). N ≥ 9. * *p* < 0.001 vs. MHV M expressed with MYO5B+D using Student’s *t*-test. ** *p* < 0.0001 vs. MHV M expressed with MYO5B+D using Student’s *t*-test. (**B**) M-GFP proteins from PEDV, MERS and SARS-CoV-2 co-expressed with either MYO5B+D or MYO5BΔD (MYO5B-D). ** *p* < 0.0001 vs. M protein expressed with MYO5B+D using Student’s *t*-test. N ≥ 9. (**C**) Colocalization of Cherry- MYO5B+D with M-GFP chimeras for PEDV, MERS and SARS-CoV-2, compared with E to K mutants of each M protein. N ≥ 6. * *p* < 0.001 vs. between mutant and wild type M protein expressed with MYO5B+D using Student’s t-test. ** *p* < 0.0001 vs. between mutant and wild type M protein expressed with MYO5B+D using Student’s *t*-test. Analysis was performed on a minimum of three fields from at least three separate experiments.

**Figure 3 cells-13-00126-f003:**
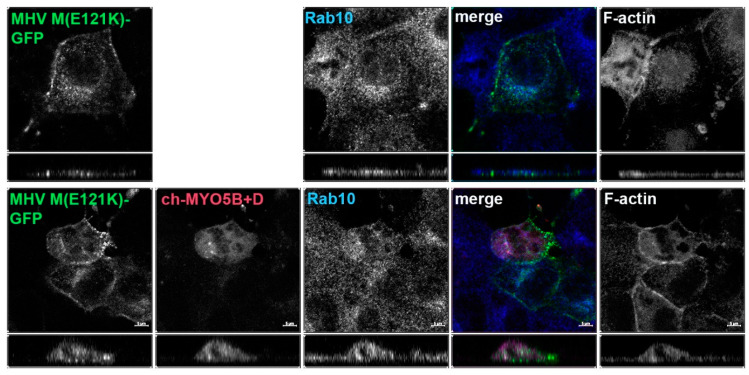
The E121K point mutant in MHV M cytoplasmic tail blocks co-localization with co-expressed Cherry- MYO5B+D. MHV M(E121K)-GFP mutant was expressed alone or with Cherry- MYO5B+D in MDCK cells. All cells were immunostained for endogenous Rab10 and F-actin (Phalloidin). Labels on the individual panels indicate the color used to produce the three-color merged image. The panels to the right of the merged panel were not used to produce the merged image. *Z* axis projections are shown below X-Y slice images, with the merged overlap at the right. Bar = 5 µm. Results are representative of four individual experiments.

**Figure 4 cells-13-00126-f004:**
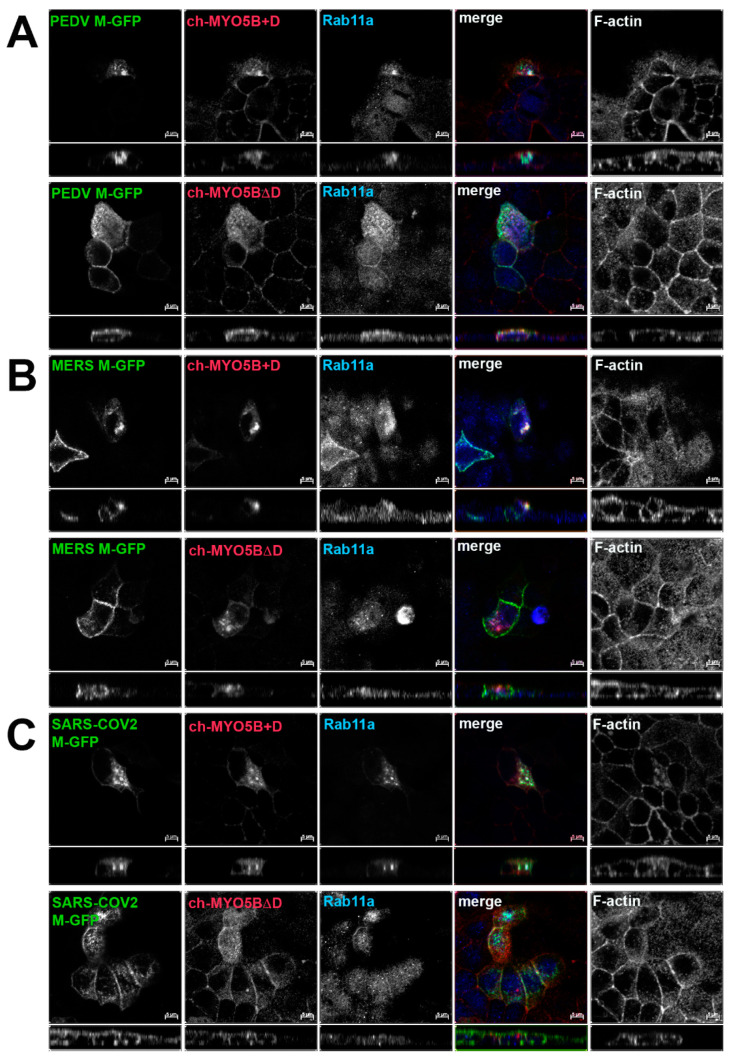
Co-localization of coronavirus M proteins with co-expressed MYO5B+D. GFP chimeras of (**A**) PEDV, (**B**) MERS and (**C**) SARS-CoV-2 M proteins were expressed with either Cherry- MYO5B+D or Cherry-MYO5BΔD in MDCK cells. Cells were immunostained for endogenous Rab11a and F-Actin (Phalloidin). Labels on the individual panels indicate the color used to produce the three-color merged image. The panels to the right of the merged panel were not used to produce the merged image. *Z* axis projections are shown below X-Y slice images, with the merged overlap at the right. Bar = 5 µm. Results are representative of four individual experiments.

**Figure 5 cells-13-00126-f005:**
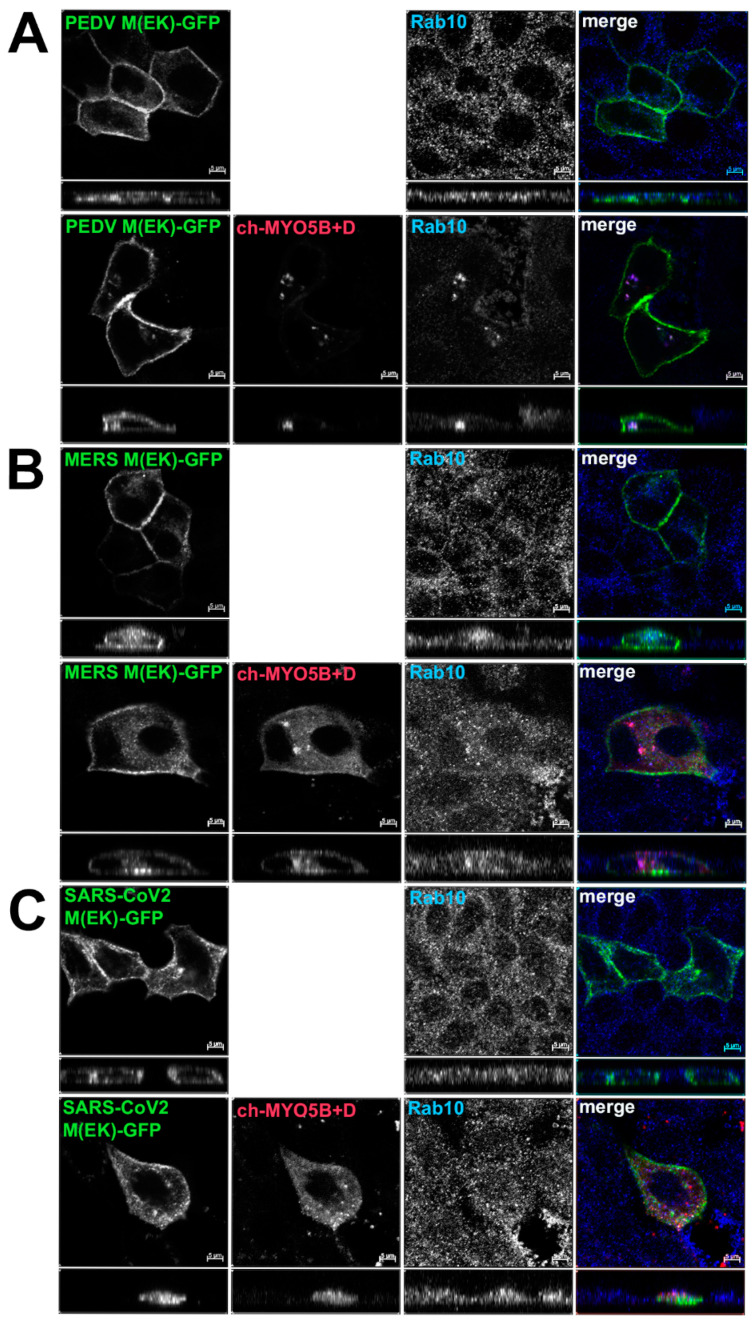
Loss of co-localization with Cherry- MYO5B+D with point mutations in coronavirus M proteins. GFP chimeras of (**A**) PEDV(E115K), (**B**) MERS(E114K) and (**C**) SARS-CoV-2(E115K) M protein mutants were expressed either alone or with Cherry- MYO5B+D in MDCK cells. Cells were immunostained for endogenous Rab10. Labels on the individual panels indicate the color used to produce the merged image. *Z* axis projections are shown below X-Y slice images, with the merged overlap at the right. Bar = 5 µm. Results are representative of four individual experiments.

**Figure 6 cells-13-00126-f006:**
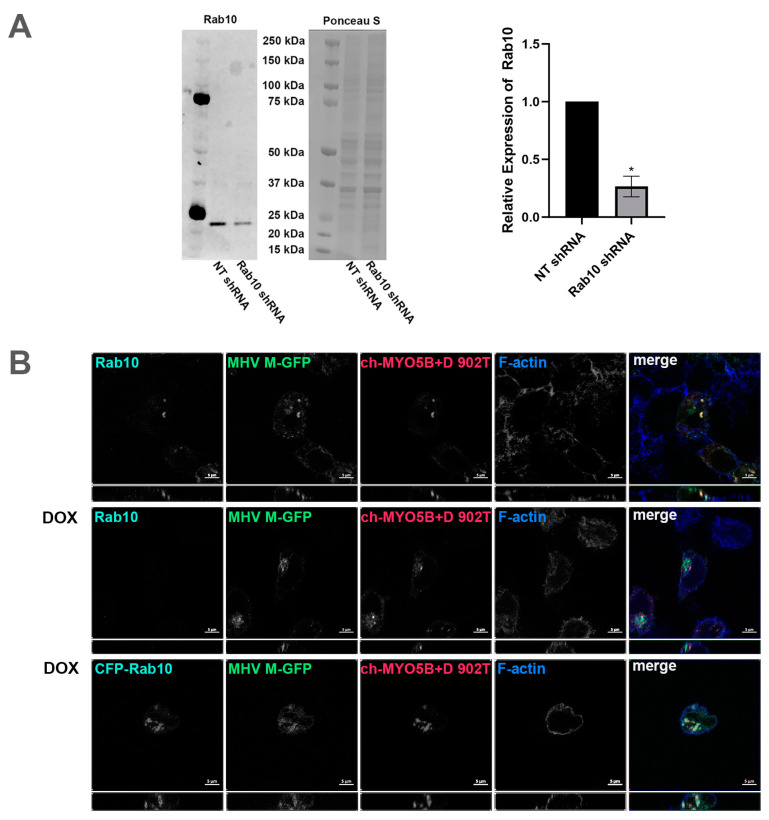
Knockdown of Rab10 causes loss of colocalization of MHV M protein with MYO5B+D. (**A**) Rab10 expression in protein lysates from A549 pIND pLKO control shRNA and A549 pIND pLKO Rab10 shRNA cells after 72 h of daily induction with doxycycline was measured by Western blot. The intensities of Rab10 bands were measured using ImageJ.JS and normalized to the corresponding Ponceau S intensities. The mean intensities of the control bands were set to 1. Across four replicate experiments, the mean intensity of the Rab10 band in the Rab10 shRNA cells was 0.265 ± 0.0444 standard error. * *p*  <  0.0001 compared with NT shRNA controls across four experiments using Student’s *t*-test. (**B**) The inducible Rab10 KD A549 cells were dual transfected with mCherry- MYO5B+D tail (902T; amino acids 902–1848) and MHV M-GFP. A third set was also transfected with CFP-Rab10, the third row. This set and another, row two, were incubated with doxycline (DOX) for 48–72 h, with fresh doxycycline added every 24 h. Cells were co-immunostained for endogenous Rab10 and for CFP-Rab10 rescue, where Cer-Rab10 fluorescence is shown, the third row. Cells were also stained for Phalloidin (F-actin). Labels on the individual panels indicates the color used to produce the three-color merged image. Images are representative of three separate experiments. Note that Rab10 KD caused a loss of co-localization of MHV M-GFP with mCherry- MYO5B+D tail.

**Figure 7 cells-13-00126-f007:**
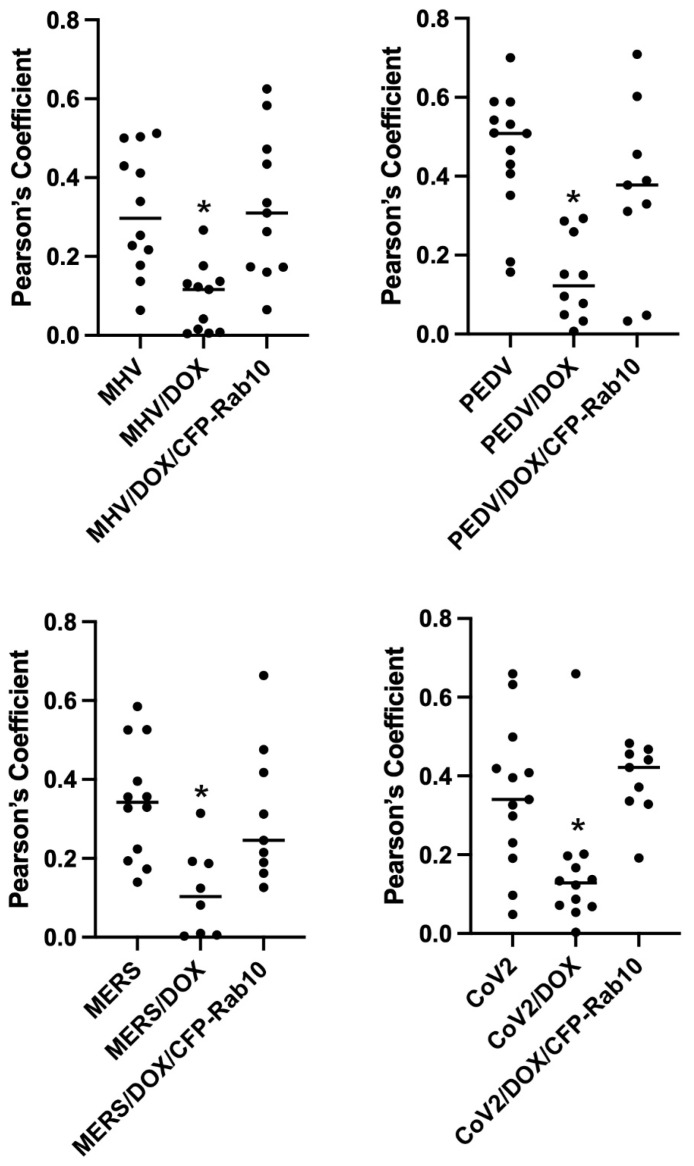
Knockdown of Rab10 expression in A549 cells significantly decreases co-localization of Coronavirus M proteins co-expressed Cherry- MYO5B+D tail. A549 lung cells were transfected with MHV M-GFP, PEDV M-GFP, MERS M-GFP or SARS-CoV-2 M-GFP, along with Cherry- MYO5B+D 902 tail (902T), in non-targeting (NT) wild type cells, Rab10 knockdown (KD) cells and Rab10 knockdown cells with rescue with re-expression of CFP-Rab10. Pearson’s coefficients were calculated in Z stacks. * *p* < 0.01 using Tukey’s test compared with non-doxycycline treated cells.

**Table 1 cells-13-00126-t001:** Split-ubiquitin assay of M protein interactions with MYO5B(ABCDE). Yeast growth was rated on a scale of 0–8 in which 0 was no growth, 1 was a few colonies in the 1× spot, 2 was a solid dot in the 1× spot and 3 was a solid dot in 1× and a few colonies in the 1:10 spot. Results are the rounded averages of at least four experiments.

	DSL Empty	DSL MYO5B(ABCDE)
pCCW MHV M	0	3
pCCW SARS-CoV-2 M	0	1
pCCW MERS M	0	3
pCCW PEDV M	0	2

## Data Availability

All data are available in the figures and Appendix A in this manuscript.

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
