# Peer review of "Coronavirus M Protein Trafficking in Epithelial Cells Utilizes a Myosin Vb Splice Variant and Rab10"

_cells, 2024, doi:10.3390/cells13020126_

Round 1

Reviewer 1 Report

Comments and Suggestions for Authors

This interesting submission provides convincing evidence that the C-terminal cytoplasmic portions of coronavirus M proteins interact with a rare myosinV (Myo5B + D).  The Myo5B+D is known to interact with Rab10, providing a possible connection from M-containing vesicles to the actin cytoskeleton, for directed vesicle trafficking.  The authors therefore suggest that these interactions are meaningful in M protein transport in the organelles that participate in CoV assembly.  

While the findings are clear and potentially important, the paper comes across as incomplete with respect to its relevance in coronavirus biology.  The importance of the M protein : Myo5B + D interaction was not assessed in relation to coronavirus infection, and therefore the authors could only speculate on biological significance.  For example, the interesting comment on lines 320-322, “MYO5B + D may serve as a break on .. forward membrane protein trafficking… to allow completion of virion assembly” is very interesting but does not have any experimental support.  Therefore, the report may well provide impetus to further probe M interactions with myosin in relation to virus trafficking, but it can only suggest myosin importance on the basis of results with the nonfunctional M E121K mutants.  That M proteins with E121K changes are both nonfunctional in virus assembly and also lack MYO5B + D binding is interesting, but could there be other explanations for E121K failure to support assembly, other than lack of MYO5B + D interaction?

1.     Have the Rab10 KD A549 cells been evaluated in CoV infection studies?  Does Rab10 KD reduce susceptibility of cells to CoV assembly or egress, or reduce CoV VLP production?  Are MYO5B+D KD cells in place, and if yes, are such cells less capable in CoV assembly or egress?  Findings would promote biological significance.

2.     Do the E121K changes interfere at the virus assembly stages or at the virus egress stages?  Are the MYO5B+D and Rab10 expected to operate at CoV assembly, or at the later virus egress stages?  The discussion points seem to imply the latter, but the abstract and other parts of the paper state that E121K prevents assembly.  Some clarification would help here.

3.     Do GFP appendages on the C-terminus of M proteins change M protein subcellular localizations, independent of any association with MYO5B+D?  Or can the colocalization results be considered natural reflections of the interactions in infected cells?  If not, then the caveats of the study should be communicated in the discussion section.

4.     The split ubiquitin yeast 2 hybrid is described on lines 206-215, and the methods for this are on lines 416-437, but where are the results presented in table and/or figure form?  

Author Response

Reveiwer 1

This interesting submission provides convincing evidence that the C-terminal cytoplasmic portions of coronavirus M proteins interact with a rare myosinV (Myo5B + D).  The Myo5B+D is known to interact with Rab10, providing a possible connection from M-containing vesicles to the actin cytoskeleton, for directed vesicle trafficking.  The authors therefore suggest that these interactions are meaningful in M protein transport in the organelles that participate in CoV assembly.  

While the findings are clear and potentially important, the paper comes across as incomplete with respect to its relevance in coronavirus biology.  The importance of the M protein : Myo5B + D interaction was not assessed in relation to coronavirus infection, and therefore the authors could only speculate on biological significance.  For example, the interesting comment on lines 320-322, “MYO5B + D may serve as a break on .. forward membrane protein trafficking… to allow completion of virion assembly” is very interesting but does not have any experimental support.  Therefore, the report may well provide impetus to further probe M interactions with myosin in relation to virus trafficking, but it can only suggest myosin importance on the basis of results with the nonfunctional M E121K mutants.  That M proteins with E121K changes are both nonfunctional in virus assembly and also lack MYO5B + D binding is interesting, but could there be other explanations for E121K failure to support assembly, other than lack of MYO5B + D interaction?

The reviewer’s points are well taken, but it is presently difficult to assign MYO5B+D roles in viral assembly.  In this paper we have focused on how coronavirus M protein trafficking marks the Rab10-dependent vesicle trafficking pathway.  Many more studies will be needed to define how this may interact with viral assembly, so we have removed speculation on this from the paper. 

  1. Have the Rab10 KD A549 cells been evaluated in CoV infection studies?  Does Rab10 KD reduce susceptibility of cells to CoV assembly or egress, or reduce CoV VLP production?  Are MYO5B+D KD cells in place, and if yes, are such cells less capable in CoV assembly or egress?  Findings would promote biological significance.

While we agree with this assessment, these studies are difficult to perform, since they also require co-expression of the viral receptors in A549 cells.  We hope to be performing these studies in the future to determine Rab10 influence on viral assembly.

  1. Do the E121K changes interfere at the virus assembly stages or at the virus egress stages?  Are the MYO5B+D and Rab10 expected to operate at CoV assembly, or at the later virus egress stages?  The discussion points seem to imply the latter, but the abstract and other parts of the paper state that E121K prevents assembly.  Some clarification would help here.

As noted above, these are all important issues, but we would suggest that these are also beyond the scope of the present investigation, especially with the limitations on time for revision, and will require testing of multiple cell lines with Rab10 and also Rab11a knockdown.

  1. Do GFP appendages on the C-terminus of M proteins change M protein subcellular localizations, independent of any association with MYO5B+D?  Or can the colocalization results be considered natural reflections of the interactions in infected cells?  If not, then the caveats of the study should be communicated in the discussion section.

It is certainly possible that the GFP-amino-terminal chimera may affect some aspects of trafficking and certainly it would alter incorporation into virus particles.  However, the focus of this manuscript is to define the status of the GFP-M protein chimeras as cargoes within Rab10 trafficking systems.  These CoV-M-GFP proteins therefore represent important ligands for studying Rab10-dependent trafficking.  Similar examination of viral protein trafficking of HA and VSV-G proteins have contributed greatly to cell biology research especially in polarized cells.

  1. The split ubiquitin yeast 2 hybrid is described on lines 206-215, and the methods for this are on lines 416-437, but where are the results presented in table and/or figure form?  

We have added Table 1 summarizing these results.

Reviewer 2 Report

Comments and Suggestions for Authors

The manuscript identified the interaction of coronavirus M proteins with the specific splice variant of host myosin Vb, which was related to the interaction with Rab10, using a yeast two-hybrid screen. In general, the manuscript is well written and clearly presented.

Some comments were listed below to be addressed by the authors.

- Lanes 74-89: The last paragraph summarised results of the manuscript. All results were presented and it is long. It is expected highlight of manuscript results in the “Introduction” section.

- Fig. 6A: Ladder lanes are not shown in Western blot as presented in the original images file.

- Lanes 339-347: Are pBD-Gal(Cam), pEGFP-N1 and pmCherry-C1 commercial vectors? Add more information or references. Also add information about restriction sites used for cloning M proteins into vectors.

- Figs 2, 6A and 7: inform the type of statistical analysis performed in the Figure caption.

- What is the relation between the manuscript and the article published in 2022 (the reference is shown below)?

Goldenring JR, Lapierre LA, Roland JT, Manning E, Caldwell C, Glenn HL, Vidalain PO, Tangy F, Hogue BG, de Haan CAM. A Myosin Vb Splice Variant Regulates Coronavirus M Protein Trafficking in Polarized Epithelial Cells. FASEB J. 2022 May;36(Suppl 1):10.1096/fasebj.2022.36.S1.R2283. doi: 10.1096/fasebj.2022.36.S1.R2283.

Author Response

Reviewer 2

The manuscript identified the interaction of coronavirus M proteins with the specific splice variant of host myosin Vb, which was related to the interaction with Rab10, using a yeast two-hybrid screen. In general, the manuscript is well written and clearly presented.

Some comments were listed below to be addressed by the authors.

- Lanes 74-89: The last paragraph summarised results of the manuscript. All results were presented and it is long. It is expected highlight of manuscript results in the “Introduction” section.

We have shortened this portion of the introduction.

- Fig. 6A: Ladder lanes are not shown in Western blot as presented in the original images file.

We have added the positions of standards to the figure.

- Lanes 339-347: Are pBD-Gal(Cam), pEGFP-N1 and pmCherry-C1 commercial vectors? Add more information or references. Also add information about restriction sites used for cloning M proteins into vectors.

We have added these details to the manuscript. The pBD-Gal(Cam) is from Stratagene and the EGFP and mCherry vectors are from Clontech.  Restriction sites are now indicated for cloning into the vectors.

- Figs 2, 6A and 7: inform the type of statistical analysis performed in the Figure caption.

We have added these statistical details to the manuscript.

- What is the relation between the manuscript and the article published in 2022 (the reference is shown below)?

Goldenring JR, Lapierre LA, Roland JT, Manning E, Caldwell C, Glenn HL, Vidalain PO, Tangy F, Hogue BG, de Haan CAM. A Myosin Vb Splice Variant Regulates Coronavirus M Protein Trafficking in Polarized Epithelial Cells. FASEB J. 2022 May;36(Suppl 1):10.1096/fasebj.2022.36.S1.R2283. doi: 10.1096/fasebj.2022.36.S1.R2283.

This publication was merely a meeting abstract.

Round 2

Reviewer 1 Report

Comments and Suggestions for Authors

The submission has been improved with the clarification of key findings and reductions in speculations.